# Potential role of KRAB-ZFP binding and transcriptional states on DNA methylation of retroelements in human male germ cells

Kei Fukuda[1]*, Yoshinori Makino[2], Satoru Kaneko[3], Chikako Shimura[1], Yuki Okada[2], Kenji Ichiyanagi[4], Yoichi Shinkai[1]*

[1]Cellular Memory Laboratory, RIKEN Cluster for Pioneering Research, Wako, Japan; [2]Laboratory of Pathology and Development, Institute for Quantitative Biosciences, The University of Tokyo, Tokyo, Japan; [3]Department of Obstetrics and Gynecology, Ichikawa General Hospital, Tokyo Dental College, Ichikawa, Japan; [4]Laboratory of Genome and Epigenome Dynamics, Department of Animal Sciences, Graduate School of Bioagricultural Sciences, Nagoya University, Nagoya, Japan

*For correspondence:
kei.fukuda@riken.jp (KF);
yshinkai@riken.jp (YS)

**Competing interest:** The authors declare that no competing interests exist.

**Abstract** DNA methylation, repressive histone modifications, and PIWI-interacting RNAs are essential for controlling retroelement silencing in mammalian germ lines. Dysregulation of retroelement silencing is associated with male sterility. Although retroelement silencing mechanisms have been extensively studied in mouse germ cells, little progress has been made in humans. Here, we show that the Krüppel-associated box domain zinc finger proteins are associated with DNA methylation of retroelements in human primordial germ cells. Further, we show that the hominoid-specific retroelement SINE-VNTR-*Alus* (SVA) is subjected to transcription-directed de novo DNA methylation during human spermatogenesis. The degree of de novo DNA methylation in SVAs varies among human individuals, which confers significant inter-individual epigenetic variation in sperm. Collectively, our results highlight potential molecular mechanisms for the regulation of retroelements in human male germ cells.

## Editor's evaluation

Retrotransposons undergo massive reprogramming of their methylation states during germ cell development, but some elements are immune to this remodeling. This manuscript explores the contribution of binding motifs for KRAB-Zinc Finger Proteins (KZFPs) and position towards genes to explain the variable methylation dynamics of different retrotransposon families, namely L1, SVA and LTR12, as well as potential inter-individual variation during male germ cell development in humans, using an integrative analyses of available sequencing datasets. By bringing insights into the complex regulation of retrotransposons, it could be of particular interest to the epigenetics community.

## Introduction

Transposable elements comprise more than 40% of most extant mammalian genomes (*Lander et al., 2001*). Among these, certain types of transposable elements called retroelements, including short/long interspersed elements (SINEs/LINEs) and hominoid-specific retrotransposons SINE-VNTR-*Alus* (SVA) are active in humans and can be transposed (*Huang et al., 2012*; *Maksakova et al., 2013*; *Ostertag et al., 2003*). As retrotransposons cause genome instability, insertional mutagenesis, or

transcriptional perturbation and are often deleterious to host species (*O'Donnell and Burns, 2010*), multiple defense mechanisms have evolved against transposition. The first line of defense is transcriptional silencing of integrated retroelements by chromatin modifications, such as DNA methylation and histone H3 lysine 9 (H3K9) methylation (*Fukuda and Shinkai, 2020*; *Goodier, 2016*). Most retroelement families are bound by Krüppel-associated box domain zinc finger proteins (KRAB-ZFPs), which coevolved to recognize specific retroelement families (*Imbeault et al., 2017*; *Jacobs et al., 2014*; *Wolf et al., 2015*). KRAB-ZFPs repress retroelements by recruiting KAP1/TRIM28 (*Sripathy et al., 2006*) and other repressive epigenetic modifiers (*Schultz et al., 2002*; *Schultz et al., 2001*).

Restricting retroelements is especially important for germ cells, because only germ cells transmit genetic information to the next generation. During embryonic development, primordial germ cells (PGCs) undergo epigenetic reprogramming, characterized by DNA demethylation and global resetting of histone marks in mice and humans (*Gkountela et al., 2015*; *Guo et al., 2015*; *Kobayashi et al., 2013*; *Seisenberger et al., 2012*; *Seki et al., 2007*; *Tang et al., 2015*). A subset of young retroelements resists this global DNA demethylation event in PGCs, which may be required for retroelement silencing (*Gkountela et al., 2015*; *Guo et al., 2015*; *Kobayashi et al., 2013*; *Seisenberger et al., 2012*; *Seki et al., 2007*; *Tang et al., 2015*). H3K9 trimethylation mediated by SETDB1 is enriched in DNA demethylation-resistant retroelements in mouse PGCs (*Liu et al., 2014*). As SETDB1 regulates DNA methylation of a subset of retroelements (*Matsui et al., 2010*; *Rowe et al., 2013*), and it is recruited to the retroelements via interaction with KRAB-ZFPs, it has been hypothesized that SETDB1/KRAB-ZFPs may contribute to DNA demethylation resistance in PGCs.

In contrast to the extensive DNA hypomethylation in PGCs, the genomic DNA of sperm is highly methylated in both humans and mice (*Hammoud et al., 2014*; *Kobayashi et al., 2013*; *Molaro et al., 2011*; *Okae et al., 2014*). Retroelements are also subjected to de novo DNA methylation during spermatogenesis in mice via the PIWI/piRNA pathway (*Aravin et al., 2008*; *Inoue et al., 2017*). Epigenetic alterations in retroelements and dysfunction of retroelement silencing pathways in male germ cells are associated with male sterility linked to azoospermia (*Aravin et al., 2007*; *Bourc'his and Bestor, 2004*; *Carmell et al., 2007*; *Heyn et al., 2012*; *Urdinguio et al., 2015*). In addition, epigenetic alterations of retroelements in male germ cells can be potentially transmitted to the next generation with phenotypic consequences (*Daxinger et al., 2016*; *Rakyan et al., 2003*). Therefore, deciphering the regulatory mechanisms of retroelements in germ cells contributes to the understanding of sterility and transgenerational epigenetic inheritance. Extensive studies have been conducted to understand DNA methylation mechanisms in mouse spermatogenesis; however, limited progress has been achieved in humans.

In this study, we aimed to clarify the regulatory mechanisms of DNA methylation of retroelements in human germ cells and performed an integrative analysis using three sets of previously reported data, which included whole-genome bisulfite sequencing (WGBS) data for human PGCs (hPGCs) and sperm, the transcriptome of human male germ cells, and comprehensive human KRAB-ZFPs ChIP-exo data.

## Results

### Transposable elements showing DNA demethylation resistance in hPGCs

To learn more about the factors that contribute to DNA demethylation resistance in hPGCs, we reanalyzed publicly available WGBS data for male hPGCs (*Guo et al., 2015*). The global erasure of DNA methylation is mostly complete at 19 weeks of gestation (*Figure 1A*), therefore, we analyzed the DNA methylation status of full-length transposable elements (a copy whose length is 90% or more of the length of the consensus sequence of each subtype, listed in *Supplementary file 1*) in male hPGCs at 19 weeks of gestation to identify retroelements that showed resistance to demethylation. We generally focused on the retroelement types that had been analyzed for the DNA methylation status more than 30 copies. Among the retroelements we analyzed, the primate-specific retroelement families L1PA, SVA, and LTR12 showed high levels of DNA methylation (*Figure 1B*). In the SVA family, SVA_A, which emerged 13–14 million years ago (Mya) and is the oldest SVA type, showed the highest DNA methylation levels relative to other SVA types. This includes the currently active SVA_E/F (*Figure 1C*). In the L1 family, L1PA3–5, which emerged 12–20 Mya and is moderately young, showed

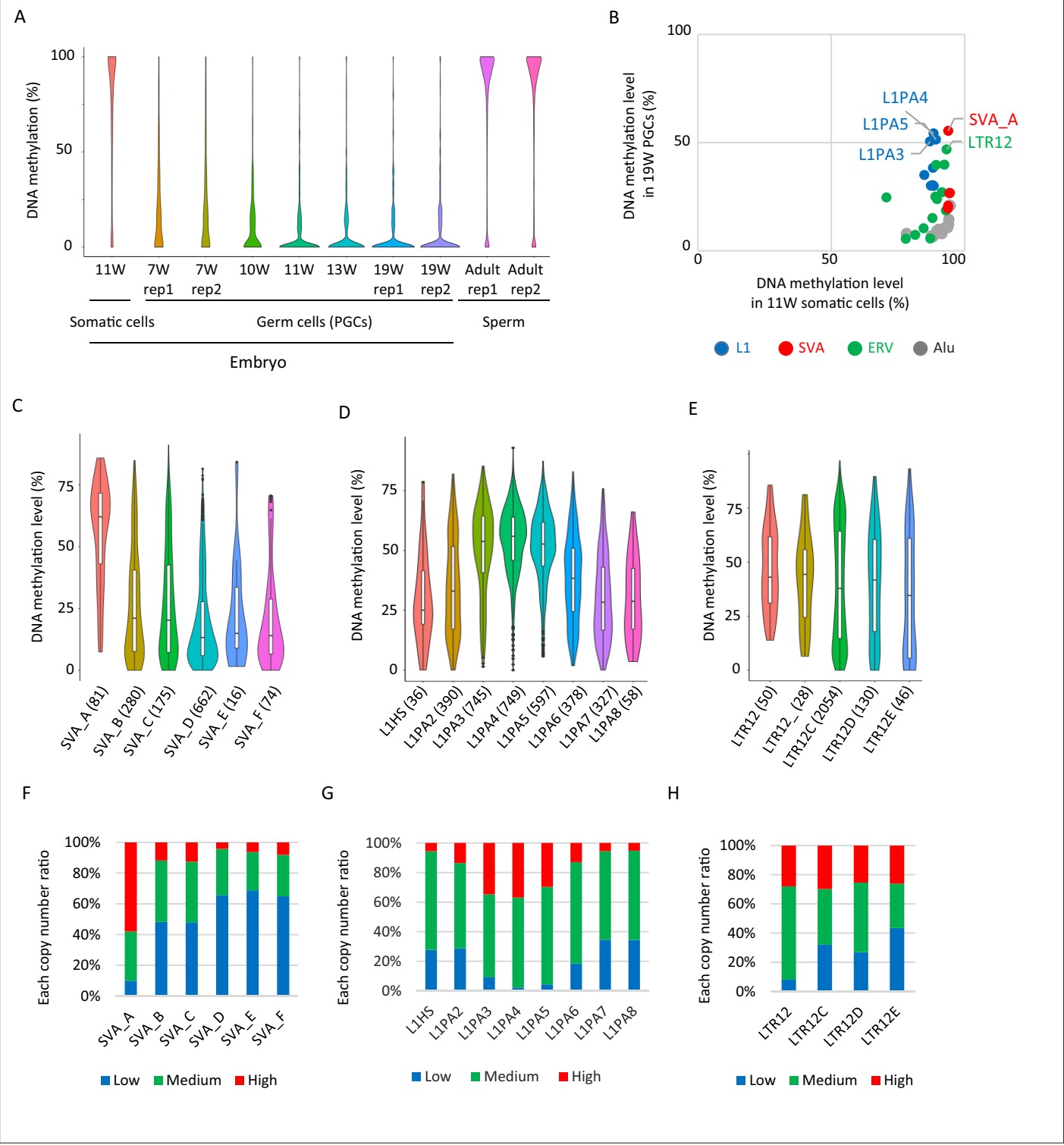

**Figure 1.** Retroelements showing DNA demethylation resistance. (**A**) Violin plots showing DNA methylation levels of each CpG site during human male germ-cell development. DNA demethylation was almost completed at 19 weeks of gestation. (**B**) Scatter plots showing average DNA methylation level of each retroelement type between somatic cells and male human primordial germ cells (hPGCs) at 19 weeks of gestation. Only full-length copies were used for this analysis, and retroelement types with ≧30 full-length copies were shown. Each plot was colored according to its retroelement family (red: SVA, blue: L1, green: LTR, gray: Alu). (**C–E**) Violin plots showing DNA methylation level of each retroelement type in hPGCs at 19 weeks of gestation. p-Value was calculated by Tukey's test and was described in *Supplementary file 2*. The number in parentheses was analyzed copy number. (**F–H**) Bar

*Figure 1 continued on next page*

*Figure 1 continued*

graphs showing the fraction of 'low', 'medium', and 'high' methylated class of each retroelement type in male hPGCs at 19 weeks of gestation. The retroelement copies used in these figures were same as those in (**C-E**).

The online version of this article includes the following source data and figure supplement(s) for figure 1:

**Source data 1.** Raw data of graphs in *Figure 1*.

**Figure supplement 1.** Mappability of whole-genome bisulfite sequencing (WGBS) reads on transposable elements.

higher methylation levels than the older (L1PA5–8) and younger L1 types, including the currently active L1 (L1HS) (*Figure 1D*). LTR12 (also known as HERV9 LTR) is not currently active and all LTR12 types are highly methylated (*Figure 1E*). Therefore, it appears that young but inactive L1PA, SVA, and LTR12 types are resistant to DNA demethylation in hPGCs. Because 100 bp short-read NGS data did not map efficiently onto the currently active L1 transposon, L1HS (*Figure 1—figure supplement 1A*), and DNA methylation of only about 10% of full-length L1HS copies could be analyzed (*Figure 1— figure supplement 1B*), it is possible that a subset of L1HS is resistant to DNA demethylation. Epigenome analysis using long-read sequence technology, such as nanopore sequencing, may provide an answer to this question (*Ewing et al., 2020*). Even though some retroelement types showed relatively high DNA methylation levels in hPGCs, the DNA methylation levels of each retroelement type were highly variable among full-length copies (*Figure 1C–E*), which prompted us to try to identify potential DNA sequences required for DNA demethylation resistance. To this end, we classified each retroelement copy according to DNA methylation levels as follows: low <20%, 20% ≤ medium < 60%, and high ≥60%. Using this classification, we determined that both the 'high' and 'low' classes of copies exist in highly methylated retroelement types in hPGCs, such as SVA_A, L1PA3, and LTR12C (*Figure 1F–H*).

## The presence of ZNF28- and ZNF257-binding motifs are correlated with demethylation resistance in SVA_A

KRAB-ZFPs are important factors for retroelement silencing. Their activity is mediated by the recruitment of KAP1 and SETDB1, which induces retroelement DNA methylation (*Matsui et al., 2010*). To investigate whether KRAB-ZFPs could be involved in the DNA demethylation resistance of SVAs, we reanalyzed the binding peak data of 250 KRAB-ZFPs identified by ChIP-exo using exogenously tagged KRAB-ZFPs in human HEK293T cells (*Helleboid et al., 2019*; *Imbeault et al., 2017*). We observed that the ZNF257 and ZNF28 peaks overlapped more frequently with highly methylated SVA_A copies than with lowly methylated copies (*Figure 2A*). Because peaks of ZNF611 and ZNF91, which interact with SVAs in human embryonic stem cells (hESCs) (*Haring et al., 2021*; *Jacobs et al., 2014*), were observed in both lowly and highly methylated SVA_A copies (*Figure 2A*), it is unlikely that these two KRAB-ZNPs contribute to the differences in DNA methylation among SVA_A copies. Of the 'high' SVA_A elements, 63.8% and 44.7% were bound by ZNF257 or ZNF28, respectively. However, no 'low' SVA_A showed binding (*Figure 2A*), and both 'high' and 'medium' SVA_A copies significantly overlapped with the ZNF257- or ZNF28-binding peaks (*Figure 2—figure supplement 1A*). The frequency of overlap with the ZNF257/28 peaks and the enrichment of ZNF257/28 in SVA_A were positively correlated with DNA methylation (*Figure 2B and C*), and both ZNF257 and ZNF28 showed the highest enrichment of SVA_A when SVA family members were compared (*Figure 2D*).

It is possible that the correlation between the DNA demethylation resistance of SVA_A and the binding potential of specific KRAB-ZNFs based on ChIP-exo mapping data in HEK293T cells could result from differences in read mappability. To determine the likelihood of this, we calculated the mappability of each transposon copy by virtually creating reads from the retroelements and mapping them onto the genome. Although highly methylated SVA_A copies showed greater mappability than those that were lowly methylated (*Figure 2—figure supplement 1B*), the correlation between SVA_A DNA methylation levels and enrichment for ZNF28/257 was observed even when only SVA_A copies with similar mappability (50–70%) were used for analysis (*Figure 2—figure supplement 1C*). Therefore, we concluded that the enrichment of ZNF28/257 in SVA_A in HEK293T cells is correlated with SVA_A DNA methylation levels in hPGCs.

The SVA element has a region containing variable-number tandem repeats (VNTRs) in the middle segment. SVA_A contains one type of VNTR (VNTR1), whereas the other SVA classes possess two

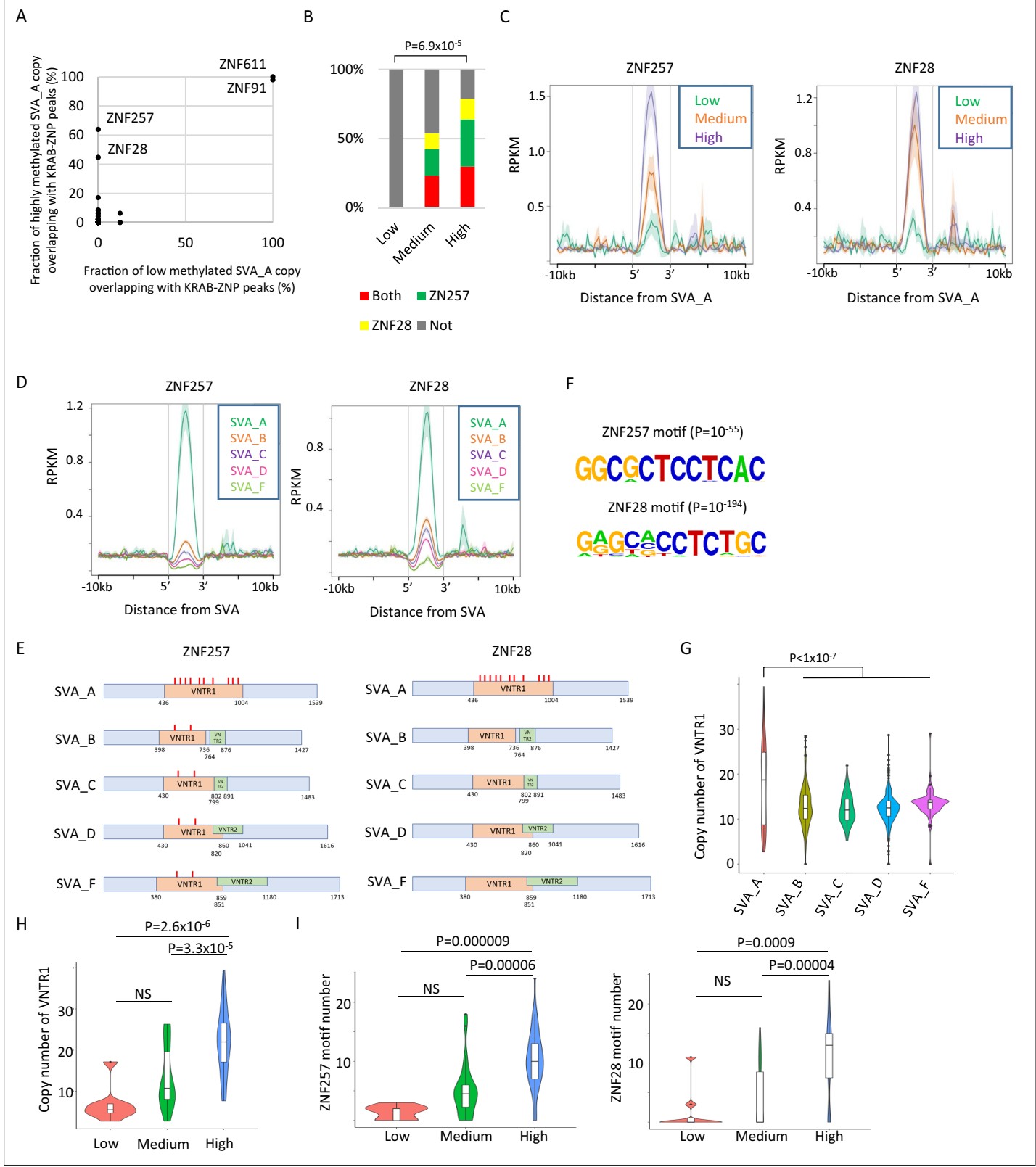

**Figure 2.** Identification of Krüppel-associated box domain zinc finger proteins (KRAB-ZFPs) associated with DNA demethylation resistance in SINE-VNTR-*Alus* (SVAs). (**A**) Scatter plots showing the fraction of low-methylated or highly methylated SVA_A copies which overlaps of KRAB-ZFP peaks. ZNF257 and ZNF28 peaks were more frequently overlapped with 'high' methylated SVA_A than 'low' methylated SVA_A. For this analysis, publicly available ChIP-exo data from 250 human KRAB-ZFPs in HEK293T cells were used. (**B**) Bar graphs showing the fraction of SVA_A copies with ZNF257 and

*Figure 2 continued on next page*

*Figure 2 continued*

ZNF28 peaks. SVA_A copies were classified by DNA methylation levels in 19 W human primordial germ cells (hPGCs) (N = 8, 26, 47 in low, medium, and high, respectively). p-Value was calculated by chi-square test. (**C**) Enrichment of ZNF257 and ZNF28 on SVA_A classified by DNA methylation levels in male hPGCs at 19 weeks of gestation. (**D**) Enrichment of ZNF257 and ZNF28 on each SVA subtype. (**E**) Position of ZNF257- and ZNF28-binding motifs along SVA consensus sequences. VNTR1 and VNTR2 are composed of multiple copy number of tandem repeats, and the copy number of these number tandem repeats (VNTRs) is highly variable among SVA copies. Both ZNF257- and ZNF28-binding motifs were found within VNTR1 of SVAs. (**F**) Sequence logo of ZNF257- and ZNF28-binding motifs. (**G**) Violin plots showing copy number of VNTR1 of each SVA subtype. (**H**) Violin plots showing VNTR1 copy number of SVA_A classified by its DNA methylation status in male hPGCs at 19 weeks of gestation. (**I**) Violin plots showing the number of ZNF257 and ZNF28 motifs in SVA_A classified by DNA methylation status in male hPGCs at 19 weeks of gestation. p-Value was calculated by Tukey's test.

The online version of this article includes the following source data and figure supplement(s) for figure 2:

**Source data 1.** Raw data of graphs in *Figure 2*.

**Figure supplement 1.** Correlation between ZNF257/ZNF28-binding motifs and DNA methylation levels in SVA_A copies.

**Figure supplement 2.** Expression of Krüppel-associated box domain zinc finger proteins (KRAB-ZFP) genes in somatic cells and human primordial germ cells (hPGCs).

types of VNTRs (VNTR1 and VNTR2) (*Figure 2E*). The ZNF257- and ZNF28-binding motifs, which were predicted by HOMER (*Heinz et al., 2010*; *Figure 2F*), are in VNTR1 (*Figure 2E*, *Figure 2—figure supplement 1D*). The number of ZNF257- and ZNF28-binding motifs within SVAs was the highest in SVA_A (*Figure 2E*) and was most strongly correlated with the copy number of VNTR1 in SVA_A out of all the SVA classes (*Figure 2G*). The VNTR1 copy number was also highly variable among SVA_A copies (*Figure 2G*), and DNA methylation of SVA_A was positively correlated with the VNTR1 copy number (*Figure 2H*, *Figure 2—figure supplement 1E*) and the number of ZNF257/28 motifs (*Figure 2I*, *Figure 2—figure supplement 1F*). We also confirmed that DNA methylation levels within the VNTR were correlated with ZNF257 or ZNF28 association (*Figure 2—figure supplement 1G, H*). These results indicate that a high number of ZNF257- and ZNF28-binding motifs within the VNTR increases the enrichment of KRAB-ZFPs. This might contribute to maintaining SVA_A DNA methylation during hPGC development. We confirmed the RNA expression of *ZNF257* and *ZNF28* in hPGCs by reanalysis of single-cell RNA-seq data from hPGCs and neighboring somatic cells (*Guo et al., 2015*; *Figure 2—figure supplement 2A, B*). However, there was no direct evidence for ZNF28/257 protein expression and its binding to SVAs in hPGCs, which warrants further studies.

## The presence of the ZNF649-binding motif is correlated with demethylation resistance in L1s

We also analyzed the correlation between KRAB-ZFP-binding motifs and the DNA methylation status of L1s and LTR12s in hPGCs. Consistent with previous reports that ZNF649 and ZNF93 bind L1s (*Cosby et al., 2019*; *Jacobs et al., 2014*), ZNF649 and ZNF93 peaks were frequently found in L1PA2–6 and L1PA3–6, respectively (*Figure 3A*), and these two KRAB-ZFPs were enriched at the 5′ terminus of the L1 sequences (*Figure 3B*). The frequency of L1 copies overlapping with ZNF649 and ZNF93 peaks was correlated with the DNA methylation levels of L1s in hPGCs (*Figure 3C*, *Figure 3—figure supplement 1A*). Because read mappability in L1 (L1PA4) was similar across the different DNA methylation groups (*Figure 3—figure supplement 1B*), ZNF649 and ZNF93 are candidate factors for the DNA demethylation resistance of these L1s. As was the case for SVA_A, the presence of ZNF649- or ZNF93-binding motifs (*Figure 3D*) was also correlated with DNA methylation levels (*Figure 3E*).

Reanalysis of single-cell RNA-seq data for hPGCs and neighboring somatic cells (*Guo et al., 2015*) showed that both ZNF649 and ZNF93 were expressed more in hPGCs than in neighboring somatic cells (*Figure 2—figure supplement 2A, B*). Because the correlation between the presence of binding motifs and DNA methylation levels was stronger in ZNF649 than in ZNF93 (*Figure 3E*), we investigated ZNF649 in more detail. The ZNF649-binding motif was located at the 5′ UTR of L1s (*Figure 3F*), consistent with the enrichment of ZNF649 in the 5′ UTR (*Figure 3B*). The enrichment of ZNF649 in L1s was decreased in L1PA2 and abrogated in L1HS (*Figure 3B*). Along with the decreased ZNF649 enrichment, a base substitution at the fifth position of the ZNF649-binding site was observed in the consensus sequences of L1HS (*Figure 3F*). Because the fifth position of the ZNF649-binding site (T) is conserved in highly methylated L1 copies (*Figure 3G*), a T in this position may be required for ZNF649 to bind to L1. Although highly methylated L1 copies had two mismatches within the ZNF649-binding motif, one at the third position (T→G) and one at the sixth position (A→T) (*Figure 3G*), a minor fraction

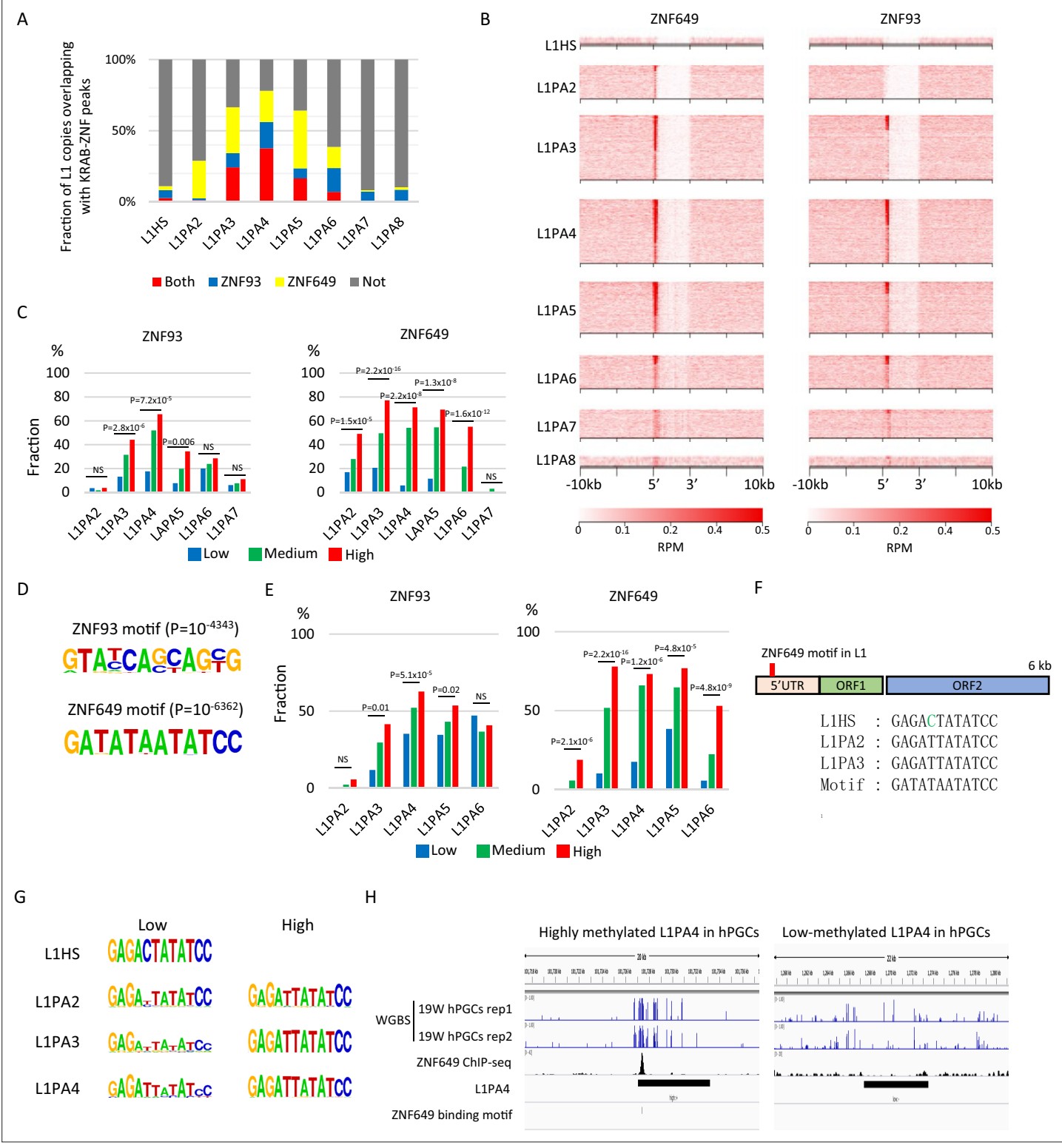

**Figure 3.** Identification of Krüppel-associated box domain zinc finger proteins (KRAB-ZFPs) associated with DNA demethylation resistance in L1. (**A**) Bar graphs showing the fraction of full-length L1 copies with ZNF93 and ZNF649 peaks. (**B**) Heatmaps showing enrichment of ZNF649 and ZNF93 along full-length L1 copies. ZNF649 binds 5′ regions of L1PA2–PA8, while ZNF93 binds the 5′ regions of L1PA3–PA8. (**C**) Bar graphs showing the fraction of L1 copies with ZNF93 and ZNF649 peaks. L1 copies were classified by their type and DNA methylation levels in male human primordial germ cells (hPGCs) at 19 weeks of gestation. (**D**) Sequence logo of ZNF93- and ZNF649-binding motifs. (**E**) Bar graphs showing the fraction of L1 copies with ZNF93- and ZNF649-binding motifs. The presence of ZNF93- and ZNF649-binding motifs was correlated with higher DNA methylation of L1 in male hPGCs at

*Figure 3 continued on next page*

*Figure 3 continued*

19 weeks of gestation (L1PA2 and -PA6 were not significant for ZNF93). p-Value was calculated by Hypothesis Testing for the Difference in the Population Proportions using a function of prop.test by R. (**F**) Comparison of sequences of ZNF649-binding sites among L1 types. L1HS lost the ZNF649 motif by a base substitution. (**G**) Comparison of sequences at ZNF649-binding sites between low- and high-methylated L1. (**H**) Representative view of correlation between DNA methylation of L1PA4 in hPGCs and ZNF649-binding peak.

The online version of this article includes the following source data and figure supplement(s) for figure 3:

**Source data 1.** Raw data of graphs in *Figure 3*.

**Figure supplement 1.** Enrichment of ZNF649/93 on L1PA4.

**Figure supplement 2.** ZNF850 association is correlated with DNA demethylation resistance of LTR12 family in human primordial germ cells (hPGCs).

of the ZNF649-binding motif had the same base composition at these sites (*Figure 3D*). Thus, these two mismatches may not abrogate ZNF649 binding. We also confirmed high DNA methylation in the ZNF649-binding motifs at individual loci (*Figure 3H*).

## The presence of the ZNF850-binding motif is correlated with demethylation resistance in LTR12C

For the LTR12C family, we found that ZNF850 more frequently overlapped with highly methylated LTR12C/D/E copies than lowly methylated ones when we analyzed the binding peak data for 250 KRAB-ZFPs (*Helleboid et al., 2019*; *Imbeault et al., 2017*; *Figure 3—figure supplement 2A, B*). We focused on LTR12C because it had the highest analyzable copy number (LTR12C: 2054; LTR12D: 130; LTR12E: 46 copies). The ZNF850-binding motif was more frequently found in highly methylated LTR12C copies than in lowly methylated copies (*Figure 3—figure supplement 2C*). Two high-confidence binding motifs (q-value < 0.01) were identified at the 5′ portion of LTR12C consensus sequences (*Figure 3—figure supplement 2D*), which was consistent with ZNF850 enrichment in the 5′ portion of LTR12C (*Figure 3—figure supplement 2E*). Lowly methylated LTR12C copies contained mismatches more frequently at the eighth and tenth positions of the first and second predicted binding sites along LTR12C, respectively (*Figure 3—figure supplement 2F*). An example of highly methylated LTR12C loci with a ZNF850 peak is shown in *Figure 3—figure supplement 2G*. These data suggest that KRAB-ZNFs prevent DNA demethylation during male germ-cell development.

## The mode of DNA methylation acquisition during spermatogenesis varies depending on retroelement type

To investigate whether lowly methylated retroelements in hPGCs acquire DNA methylation during spermatogenesis, we analyzed publicly available human sperm WGBS data from two donors (*Hammoud et al., 2014*). The two donors were of similar age (donor 1 – 32 and donor 2 – 37), and both were white Caucasians. The dynamics of DNA methylation in retroelements during spermatogenesis vary depending on retroelement type and individual characteristics. Most L1 copies acquired DNA methylation during spermatogenesis in both individuals, whereas LTR12C copies maintained their DNA methylation status in hPGCs during spermatogenesis (*Figure 4A*). A substantial difference between individuals was observed in the SVAs. The majority of SVA copies acquired DNA methylation during spermatogenesis in sperm donor 1, but not in donor 2 (*Figure 4A*). To evaluate these trends more efficiently, we classified retroelement copies based on DNA methylation levels in sperm (common high: > 60% in both donors; high and low: > 60% in donor 1 and < 20% in donor 2; common low: < 20% in both donors). The majority of lowly methylated L1 copies in hPGCs were highly methylated in sperm cells from both donors (*Figure 4B*). In contrast, most LTR12C/D copies maintained their PGC DNA methylation status during spermatogenesis (*Figure 4C*). Among the SVA types, SVA_A showed high levels of DNA methylation in both sperm donors, whereas other SVA types showed variable DNA methylation levels when both sperm donors were compared (*Figure 4D*), especially in SVA copies that had low DNA methylation levels in hPGCs (*Figure 4E*).

## The degree of DNA methylation acquisition during spermatogenesis varies among SVA copies

Although the DNA methylation status of SVAs was highly variable between the sperm donors, a subset of SVA copies acquired DNA methylation or maintained a low methylation state during

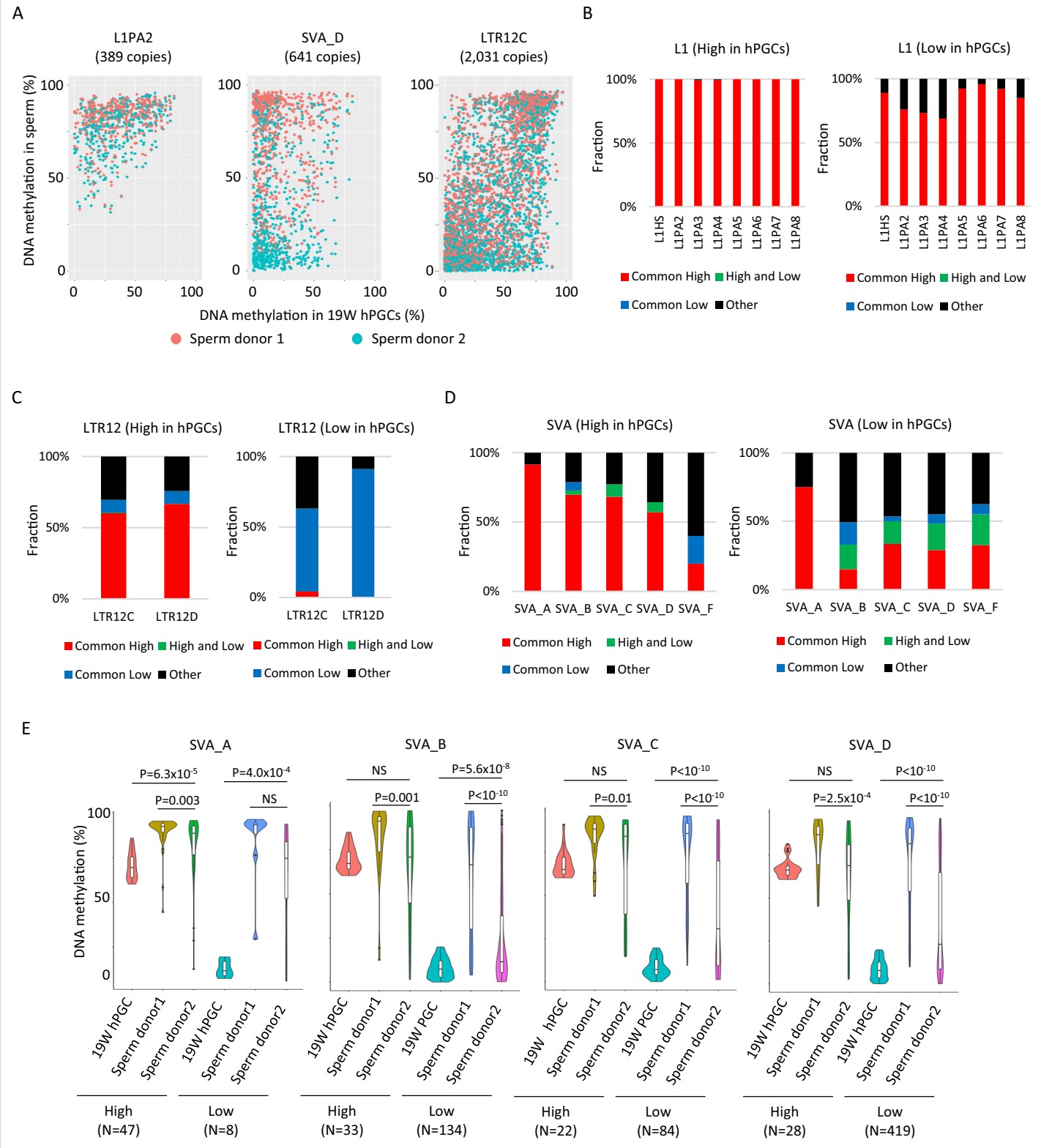

**Figure 4.** DNA methylation dynamics of retroelements during human spermatogenesis. (**A**) Scatter plots showing DNA methylation levels of each retroelement copy in male human primordial germ cells (hPGCs) at 19 weeks of gestation and sperm. Whole-genome bisulfite sequencing (WGBS) data from two sperm donors (*Hammoud et al., 2014*) were used for this analysis. Donor 1 and donor 2 were colored by orange and cyan, respectively. (**B–D**) Bar graphs showing the fraction of groups determined by DNA methylation patterns in two sperm donors in L1 (**B**), LTR12 (**C**), and SINE-VNTR-*Alus* (SVA) (**D**). 'Other' indicates groups except for common high, common low, and high and low, such as low methylated in donor 1 and mediumly

*Figure 4 continued on next page*

*Figure 4 continued*

methylated in donor 2. Bar graphs were also separated by DNA methylation levels (high or low) in male hPGCs at 19 weeks of gestation. (**E**) Violin plots showing DNA methylation levels of SVA copies in male hPGCs at 19 weeks of gestation, sperm donor 1, and sperm donor 2. The violin plots were also separated by DNA methylation levels of SVA copies in male hPGCs at 19 weeks of gestation. Although hypomethylated SVA copies in male hPGCs at 19 weeks of gestation acquired DNA methylation during spermatogenesis, the degree of DNA methylation increase was significantly different between sperm donors. p-Value was calculated by Dunnett's test.

The online version of this article includes the following source data for figure 4:

**Source data 1.** Raw data of graphs in *Figure 4*.

spermatogenesis in both sperm donors (*Figure 4D*). It is possible to get insight for mechanisms of de novo DNA methylation of SVAs during spermatogenesis by comparing SVAs that acquired DNA methylation ('low' in hPGCs and 'common high' in sperm) to SVAs that maintained hypomethylation ('low' in hPGCs and 'common low' in sperm) in both sperm donors. Phylogenetic analysis of 'common low' and 'common high' SVA copies (SVA_B and _D) showed that these two classes were not genetically separated (*Figure 5A*), indicating that the acquisition of DNA methylation in SVAs during spermatogenesis is not genetically determined.

The presence of transcription-directed retroelement silencing mechanisms, such as the PIWI/piRNA pathway (*Watanabe et al., 2018*), prompted us to investigate the correlation between the genomic distribution of SVA copies and DNA methylation. Approximately half of the SVA_B–F copies were inserted into the gene body, and most of them were in the antisense direction (*Figure 5B*). 'Common high' SVA_B–F copies were enriched in the gene body in the antisense direction, while 'common low' SVA_B–F copies were depleted from the gene body (*Figure 5B*). Reanalysis of publicly available single-cell RNA-seq data in human testes (*Sohni et al., 2019*) revealed that genes with 'common high' SVA_B–F copies in the antisense orientation showed greater expression in spermatogonial stem cells relative to genes with 'common low' (*Figure 5C*). Therefore, SVAs located in actively transcribed regions in the antisense orientation are efficiently subjected to de novo DNA methylation during spermatogenesis. The expression of genes with 'high and low' SVA_B–F copies in the antisense direction was higher in spermatogonial stem cells than the expression of other randomly extracted genes and genes with 'common low' SVA_B–F copies. However, the expression of these genes was lower than the expression of genes with 'common high' SVA_B–F copies (*Figure 5C*). Approximately half of the 'high and low' SVA_B–F copies were located in non-genic regions, but RNA-seq reads from previously reported undifferentiated spermatogonia (*Tan et al., 2020*) mapped more frequently around the non-genic 'high and low' SVA_B–F copies than the 'common low' B–F copies (*Figure 5D*). Therefore, non-genic 'high and low' SVA_B–F copies are frequently inserted in transcribed regions during spermatogenesis. These results implicate the possibility that SVAs acquire DNA methylation during DNA methylation via transcription-directed machinery, and that the effectiveness of de novo DNA methylation varies among individuals.

## SVAs are a potential source of inter-individual epigenetic variation in sperm

Inter-individual variation in DNA methylation in SVAs was also observed when another set of publicly available sperm WGBS data from three Japanese donors was analyzed (*Okae et al., 2014*; *Figure 6A*). For additional validation, we performed amplicon sequencing (amplicon-seq) of bisulfite PCR products for SVAs on sperm from five Japanese donors (*Figure 6B*). Our amplicon-seq yielded approximately 1.7–2.2 M read pairs and measured the DNA methylation level of over 90% of the full-length SVA_B–F copies (minimum read depth of CpG $\geq$ 5, analyzed CpG number $\geq$ 10) (*Figure 6C*). Again, 'high and low' SVA_B–F copies showed variations in DNA methylation among donors (*Figure 6D*). Thus, inter-individual variation in SVA methylation in sperm is a common phenomenon and is not ethnically specific.

To estimate the impact of SVAs on inter-individual epigenetic variations in sperm, we identified differentially methylated regions (DMRs) in two sperm donors, as shown in *Figures 4 and 5*; *Hammoud et al., 2014*. Although the DNA methylation profiles between the two donors were highly correlated (*Figure 6E*), 2008 regions were identified as DMRs (donor 1 < donor 2: 332, donor 2 < donor 1: 1676). Of the 1676 donor 1-specific methylated DMRs, 772 (46.1%) overlapped with SVAs (*Figure 6F*). We also observed differential DNA methylation among individuals in SVA-associated DMRs in our

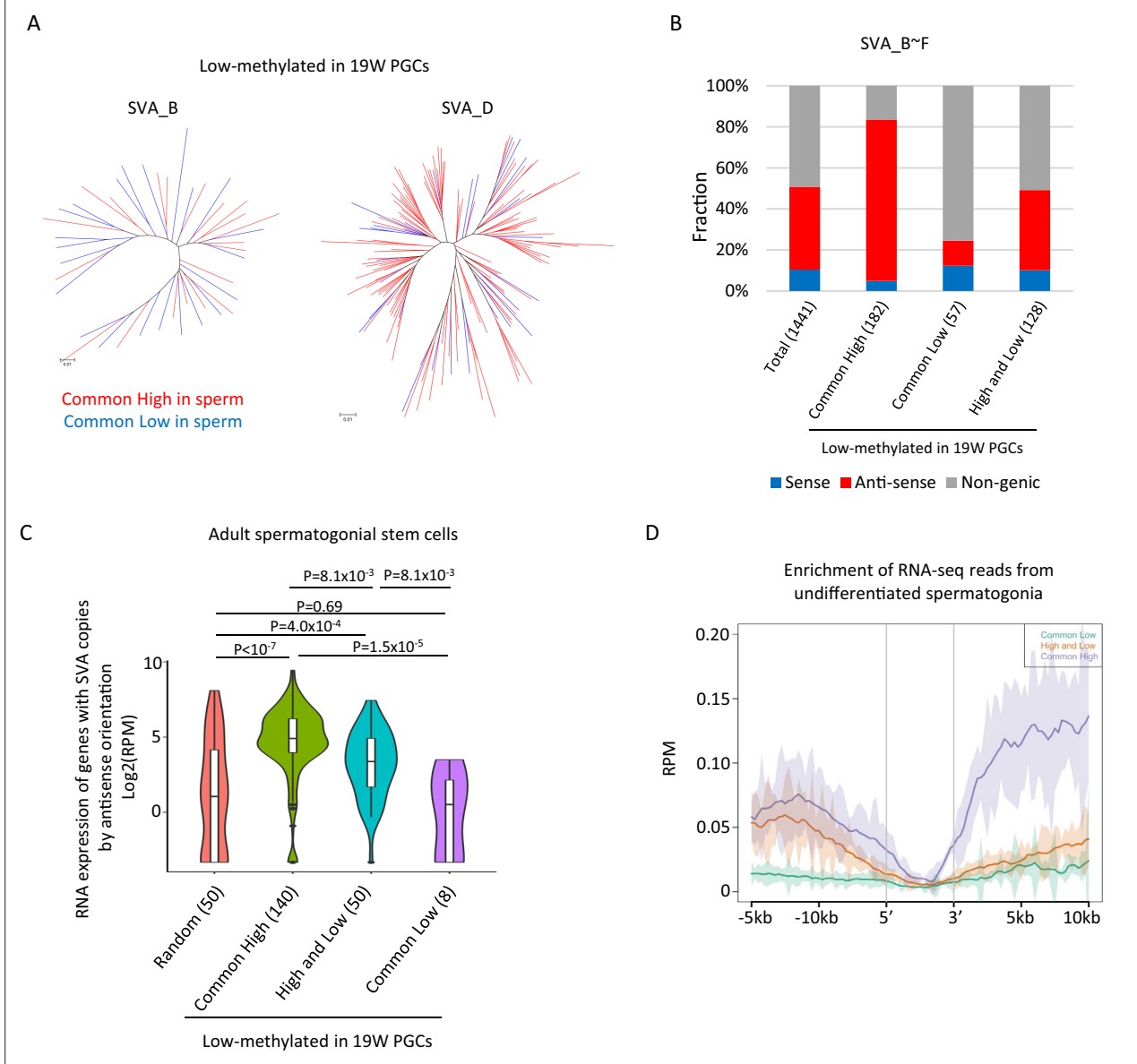

**Figure 5.** Transcription-associated regulation of DNA methylation of SINE-VNTR-*Alus* (SVA) during spermatogenesis. (**A**) Phylogenetic analysis of SVA_B (left) and SVA_D (right) copies low methylated in male human primordial germ cells (hPGCs) at 19 weeks of gestation. SVA copies highly methylated by both sperm donors were colored by red, while those hypomethylated by both sperm donors were colored by blue. (**B**) Bar graphs showing the fraction of SVA_B–F copies inserted in a gene body. SVA copies were classified by DNA methylation patterns in two sperm donors. Only low–methylated SVA copies in male hPGCs at 19 weeks of gestation were used for this analysis. The number in parentheses represents analyzed copy number. (**C**) Violin plots showing the expression of genes in adult spermatogonial stem cells 2 (*Sohni et al., 2019*). Genes were classified according to the DNA methylation status of SVAs inserted in them in the antisense direction. p-Value was calculated by Tukey's test. (**D**) Enrichment of RNA-seq reads from undifferentiated spermatogonia (*Tan et al., 2020*) around non-genic SVAs. Only low-methylated SVA copies male hPGCs at 19 weeks of gestation were used for the analysis, and SVA copies were classified by DNA methylation patterns in two sperm donors (common low, high, and low and common high).

The online version of this article includes the following source data for figure 5:

**Source data 1.** Raw data of graphs in *Figure 5*.

amplicon-seq data (*Figure 6G*). Therefore, SVAs significantly contribute to inter-individual variations in the sperm epigenome. In contrast to the inter-individual epigenetic variation of SVAs in sperm, a reanalysis of WGBS data of adult skeletal muscle from 15 individuals and of helper CD4-positive T cells from 18 individuals, which was deposited in the International Human Epigenome Consortium (IHEC) portal, showed high DNA methylation of SVAs in all individuals (*Bujold et al., 2016*; *Figure 6—figure*

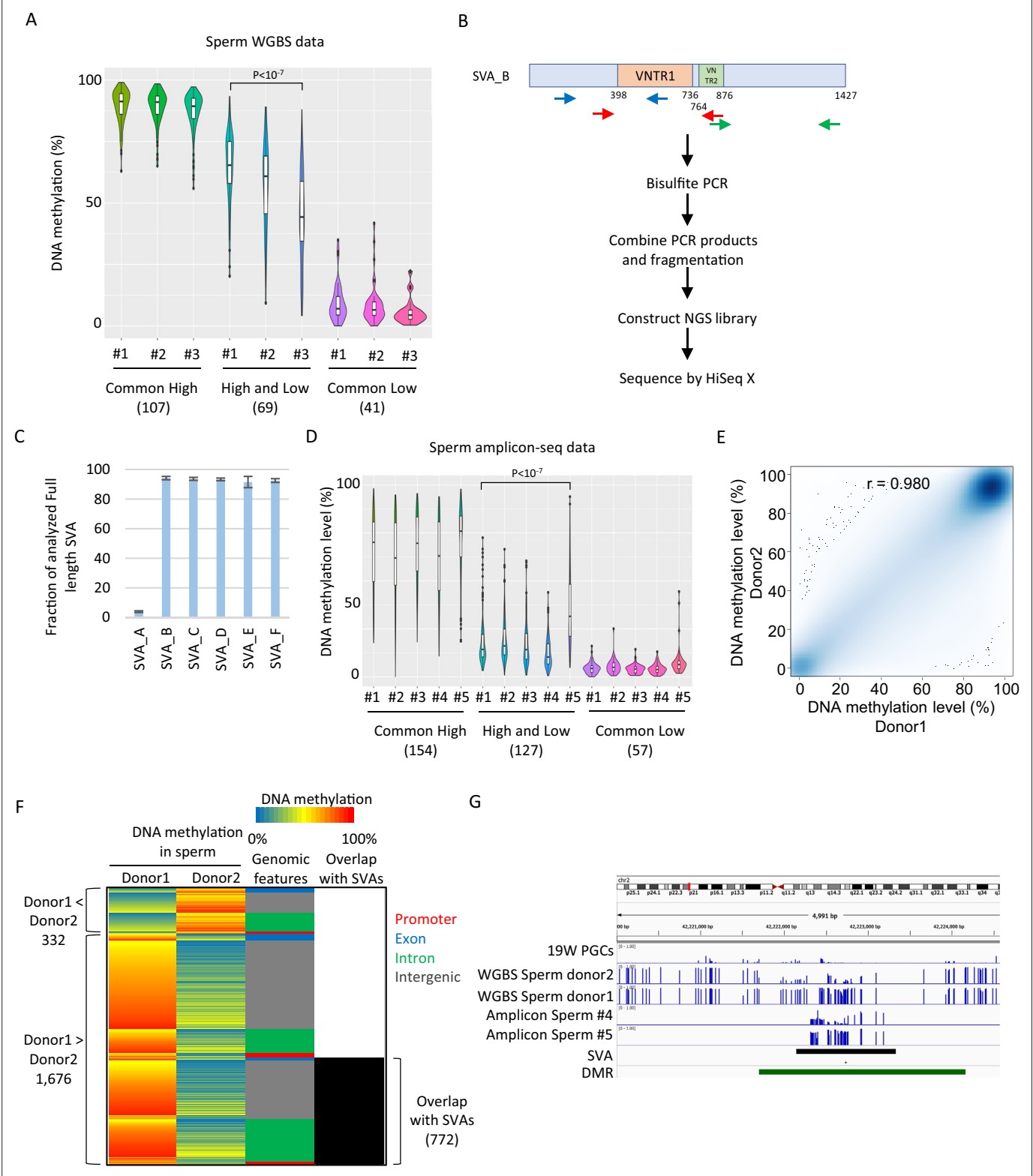

**Figure 6.** SINE-VNTR-*Alus* (SVAs) constitute a major source of inter-individual epigenetic variations in sperm. (**A**) Violin plots showing DNA methylation of SVA copies in previously reported three sperm donors (#1–#3) (***Okae et al., 2014***). Only low-methylated SVA copies in male human primordial germ cells (hPGCs) at 19 weeks of gestation were used for the analysis. SVA copies were classified by DNA methylation levels of two sperm donors from ***Hammoud et al., 2014***. Donor #1 showed significantly higher DNA methylation levels in 'high and low' SVA copies than other sperm donors. p-Value was calculated by Tukey's test. (**B**) Scheme of amplicon sequencing (amplicon-seq) for analyzing SVA methylation. (**C**) Bar plots showing the

*Figure 6 continued on next page*

*Figure 6 continued*

fraction of analyzed full-length SVA copies by amplicon-seq. (**D**) Violin plots showing DNA methylation levels of SVA copies in five sperm donors from amplicon-seq. Only low-methylated SVA copies in male hPGCs at 19 weeks of gestation were used for the analysis. SVA copies were classified by DNA methylation levels of two sperm donors from Hammoud et al. Donor #5 showed significantly higher DNA methylation levels in 'high and low' SVA copies than other sperm donors. p-Value was calculated by Tukey's test. (**E**) Scatter plot showing the DNA methylation between sperm donor 1 and sperm donor 2 from Hammoud et al. DNA methylation levels between these two donors were highly correlated. (**F**) Heatmap showing DNA methylation levels, genomic distribution, and overlap with SVAs of differentially methylated regions (DMRs). (**G**) Representative view of DMRs overlapping with SVA. Black and green boxes represent SVA and DMR, respectively.

The online version of this article includes the following source data and figure supplement(s) for figure 6:

**Source data 1.** Raw data of graphs in *Figure 6*.

**Figure supplement 1.** Inter-individual epigenetic variation in SINE-VNTR-*Alus* (SVAs) was correlated with testosterone levels in bloods.

*supplement 1A, B*). Thus, inter-individual variations in DNA methylation of SVAs in sperm are essentially canceled during development.

Finally, to investigate whether inter-individual DNA methylation variations are associated with physiological or disease conditions, we reanalyzed publicly available WGBS data from five healthy donors and six oligozoospermic patients (European Nucleotide Archive [ENA] under the accession number PRJEB34432) (*Leitão et al., 2020*). The disease condition was not associated with inter-individual DNA methylation variations in SVAs, because both healthy donors and oligozoospermic patients showed inter-individual variations of DNA methylation in 'high and low' SVA_B–F copies (*Figure 6—figure supplement 1C*). On the other hand, a comparison of various physiological conditions between highly methylated individuals and lowly methylated ones (median methylation levels of 'high and low' > 50% vs. < 50%) revealed that blood testosterone levels were significantly higher in lowly methylated individuals than in highly methylated ones (*Figure 6—figure supplement 1D*). However, prolactin, follicle stimulating hormone, luteinizing hormone (LH), sex hormone-binding globulin blood levels, and age were not significantly different between the two groups (*Figure 6—figure supplement 1D*). Although further validation of this correlation is required, DNA methylation of SVAs in sperm may be associated with physiological conditions.

## Discussion

In this study, we showed that the binding potential of KRAB-ZFPs correlates with retroelement DNA demethylation resistance in hPGCs. Furthermore, we found that de novo DNA methylation patterns in spermatogenesis vary among the L1, LTR, and SVA retroelements. In addition, we ascertained that the SVAs located in transcription-active regions in the antisense orientation are prone to methylation during spermatogenesis, which implies that the transcription-directed DNA methylation machinery might contribute to de novo DNA methylation of SVAs in male germ cells. Notably, the extent of de novo DNA methylation of SVAs in male germ cells is variable among human individuals, with SVAs being a major source of epigenetic variation in sperms.

We showed that DNA demethylation resistance in hPGCs frequently occurred in moderately young retroelements such as L1PA, SVA_A, and LTR12, but not in currently active retroelements. Because we targeted full-length transposons, our analysis was biased toward relatively young transposons. Thus, it is possible that some fragmented older transposons may also be resistant to DNA demethylation in hPGCs. A subset of LTR transposons, including LTR12, function as enhancers (*Deniz et al., 2020*). It was recently reported that LTR5s, which are Hominidae-specific LTR-type transposons and hypomethylated in hPGCs (DNA methylation levels < 10%), can function as enhancers to promote hPGC differentiation (*Xiang et al., 2022*). Therefore, in the case of LTR12C, maintaining DNA methylation might be beneficial for hPGC development because it suppresses inappropriate activation of transposon-embedded enhancer function.

In addition, KRAB-ZFP binding potentially contributed to the DNA demethylation resistance of L1s and SVAs. ZNF257/28, ZNF649/ZNF93, and ZNF850 were associated with the DNA demethylation resistance of SVAs, L1s, and LTR12Cs, respectively (*Figure 7*). In hPGCs, multiple KRAB-ZNPs were correlated with DNA demethylation resistance in the same retroelements, which may contribute to more robust or cooperative retroelement suppression. Although ZNF91 reportedly binds to the VNTR in SVAs and silences SVA expression in embryonic stem cells (ESCs) (*Haring et al., 2021*; *Jacobs*

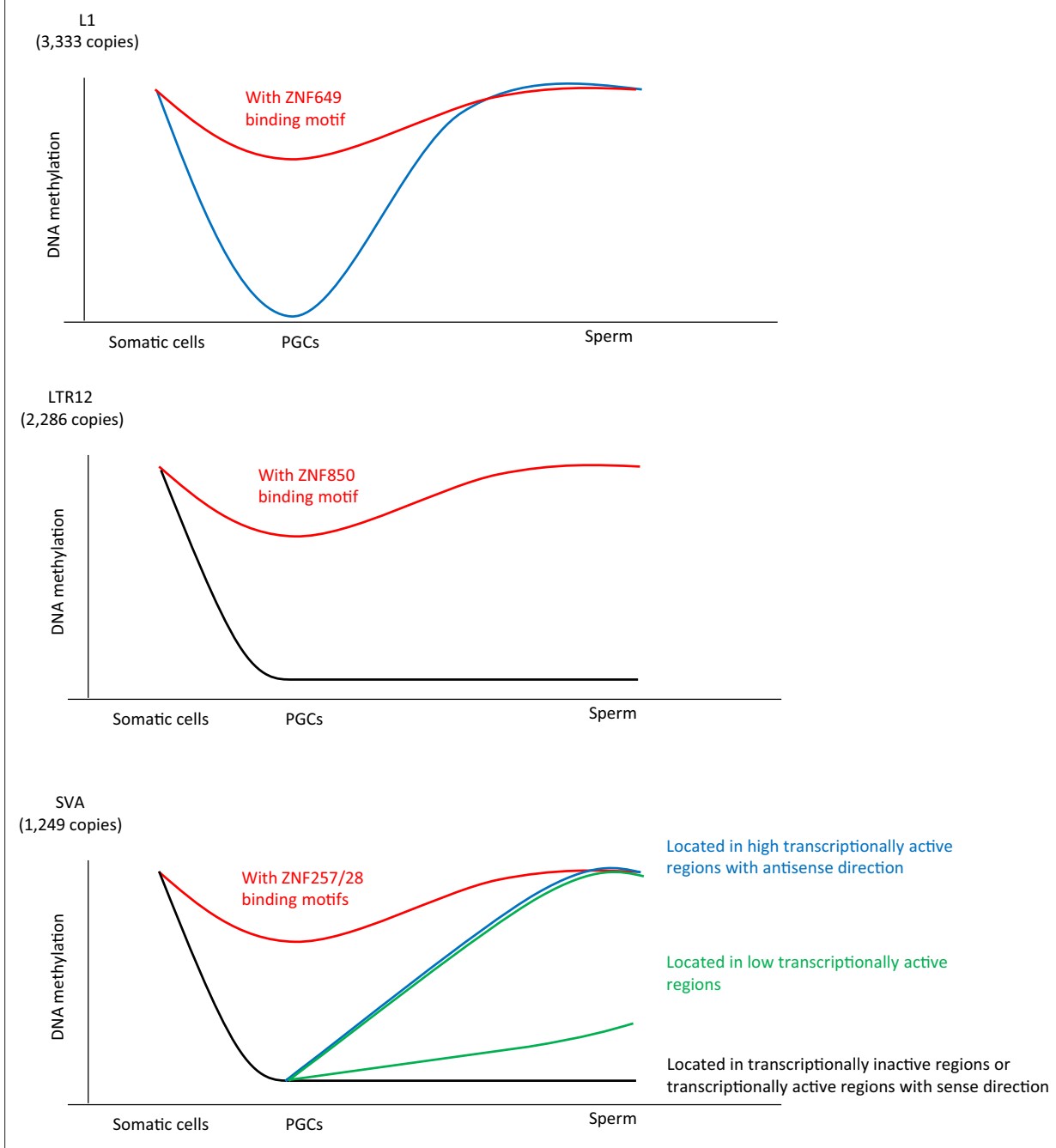

**Figure 7.** Summary of this study. Our data demonstrated an association between KRAB-ZFP-binding motifs and the DNA demethylation resistance of L1, SVA, and LTR12. ZNF649, ZNF257/28, and ZNF850 were associated with DNA demethylation resistance of L1, SVA, and LTR12C, respectively. The dynamics of DNA methylation during spermatogenesis are largely different among retroelement types. The majority of L1 copies acquired DNA methylation during spermatogenesis, whereas the DNA methylation status of LTR12 in human primordial germ cells (hPGCs) tended to be maintained during spermatogenesis. The mode of DNA methylation changes in SINE-VNTR-*Alus* (SVAs) during spermatogenesis largely differs between copies and individuals. SVA copies located in highly transcriptionally active regions acquire DNA methylation during spermatogenesis, while those located in transcriptionally inactive regions maintain a hypomethylated state during spermatogenesis. In contrast, the degree of DNA methylation in sperm in SVA copies located in low transcriptionally active regions was highly variable among the individuals. These results suggest that SVAs may be methylated by transcription-directed DNA methylation mechanisms during spermatogenesis, and their activity varies among individuals.

*et al., 2014*), the DNA demethylation resistance of SVAs did not correlate with ZNF91 binding, indicating that a different KRAB-ZFP set is used to suppress SVAs in human PGCs and ESCs. Both ZNF257 and ZNF28 bound to VNTR1 (*Figure 2E*), and high copy numbers of VNTR1 were correlated with high ZNF257 and ZNF28 enrichment and DNA methylation (*Figure 2H1*). The reduction in VNTR1 copy number after SVA_B emergence (*Figure 2G*) may have been necessary for SVAs to escape silencing mechanisms in hPGCs. The human genome encodes at least 350 KRAB-ZFPs, and not all KRAB-ZFPs were included in the ChIP-seq dataset used in this study (100 copies remained unmapped). Thus, the involvement of other KRAB-ZFPs in DNA demethylation resistance of retroelements in hPGCs is possible. Although we observed a strong correlation between KRAB-ZFPs and DNA demethylation resistance, direct evidence for this correlation remains elusive because of the limited availability of human fetal gonads and of high-specificity antibodies for KRAB-ZFPs. Because they can function as in vitro derivation systems, PGC-like cells (PGCLCs) may be a promising model for investigating the biology of PGCs. Although successful establishment of human PGCLCs has been reported (*Sasaki et al., 2015*), sufficient DNA demethylation has not been observed in human PGCLCs (*von Meyenn et al., 2016*). Thus, the currently available human PGCLCs are not suitable models for investigating the mechanisms of DNA demethylation resistance. Optimizing the derivation conditions for human PGCLCs will aid in our understanding of retroelement silencing in PGCs.

Additionally, we showed that the mode of DNA methylation acquisition during spermatogenesis was very different among retroelement types. The majority of L1 copies acquired DNA methylation during spermatogenesis, whereas LTR12 maintained its DNA methylation status in hPGCs during spermatogenesis (*Figure 7*). L1HS, which both ZNF93 and ZNF649 were unable to bind, also acquired DNA methylation during spermatogenesis (*Figure 4B*), suggesting the involvement of other factors in the de novo DNA methylation of L1 during spermatogenesis. The PIWI-piRNA pathway is responsible for the DNA methylation of L1 transposons in mouse male germ cells (*Aravin et al., 2007*; *Carmell et al., 2007*; *Kojima-Kita et al., 2016*; *Manakov et al., 2015*; *Shoji et al., 2009*). The PIWI-piRNA pathway may also be functional in humans, because mutations in genes associated with the PIWI-piRNA pathway are linked to human male infertility (*Arafat et al., 2017*; *Gu et al., 2010*). Moreover, the majority of putative piRNAs that mapped to transposons at gestational week 20 are derived from L1 (*Reznik et al., 2019*). Therefore, the PIWI-piRNA pathway is a candidate pathway for L1 silencing in human male germ cells.

Our data showed that the acquisition of DNA methylation of SVAs during spermatogenesis correlated with the inserted regions and not with the nucleotide sequence. SVAs inserted in transcriptionally active regions in the antisense direction are efficiently targeted for de novo DNA methylation during spermatogenesis. In mouse spermatogenesis, MIWI2 binds piRNAs and is recruited to the nascent transcribed regions that are complementary to piRNAs (*Watanabe et al., 2018*). Subsequently, MIWI2-interacting protein SPOCD1, which forms a complex with DNMT3A and DNMT3L, and potentially with a rodent-specific DNA methyltransferase DNMT3C (*Barau et al., 2016*), induces DNA methylation on transposons (*Zoch et al., 2020*). Therefore, one possible mechanism for the de novo DNA methylation of SVAs during spermatogenesis is that the MIWI2/SVA-derived piRNA complex targets nascent transcripts with antisense SVAs and induces DNA methylation. There are also other transcription-directed repetitive element silencing mechanisms, such as those involving the HUSH complex, which repress L1s and SVAs (*Fukuda et al., 2018*; *Liu et al., 2018*; *Robbez-Masson et al., 2018*). The HUSH complex targets young full-length L1s located within the introns of actively transcribed genes (*Fukuda and Shinkai, 2020*; *Liu et al., 2018*). In addition to the HUSH complex, efficient pericentromeric heterochromatin formation requires the transcription of pericentromeric satellite repeats, which stabilize SUV39H pericentromeric localization (*Johnson et al., 2017*; *Shirai et al., 2017*; *Velazquez Camacho et al., 2017*). Because SUV39H is also associated with retroelement silencing (*Bulut-Karslioglu et al., 2014*), both the HUSH complex and SUV39H are candidate factors associated with the transcription-directed DNA methylation of SVAs in human male germ cells. In eukaryotes, gene bodies are the most conserved targets of DNA methylation. Gene body DNA methylation levels are often correlated with transcriptional levels (*Teissandier and Bourc'his, 2017*). This is because of the interaction between the elongating RNA polymerase II and SETD2, which results in H3K36me3. H3K36me3 participates in the de novo methylation of DNA by recruiting DNMT3 enzymes via their chromatin reading PWWP domains (*Baubec et al., 2015*; *Shirane et al., 2020*). Furthermore, antisense RNAs embedded within protein-coding genes are selectively silenced

by H3K36 methyltransferase SET2 in *Saccharomyces cerevisiae* (*Venkatesh et al., 2016*). Thus, the machinery for gene body DNA methylation regulated by SETD2 is also a candidate for the de novo DNA methylation of SVAs during spermatogenesis.

The mechanism underlying inter-individual epigenetic variations in SVA in human sperm is unknown. In addition to genetic differences among individuals, both intrinsic and extrinsic environmental differences may contribute to inter-individual variations in SVAs. Our data indicate that in sperm, the degree of DNA methylation of SVAs located in genomic regions with low transcriptional activity varies among individuals. Thus, the effectiveness of transcription-directed de novo DNA methylation in male human germ cells may vary among individuals. Previous studies have shown that hypermethylation of the *PIWIL2* and *TDRD1* promoter regions, which are involved in the PIWI-piRNA pathway, is associated with abnormal DNA methylation and male infertility in humans (*Heyn et al., 2012*). Therefore, the effectiveness of the PIWI-piRNA pathway may vary among individuals and contribute to the epigenetic variation of SVAs in male germ cells. SVAs function as enhancers (*Gianfrancesco et al., 2017*), alter the chromatin state near the insertion site (*Fukuda et al., 2017*), and are associated with Fukuyama-type congenital muscular dystrophy and Lynch syndrome (*Ostertag et al., 2003*; *Payer and Burns, 2019*). Therefore, differences in SVA regulation among individuals may induce changes in gene regulation in male germ cells, alter the risk of genome instability, and affect the incidence of disease among individuals.

## Materials and methods

### Semen collection

Ejaculates were provided by patients who visited the Reproduction Center of the Ichikawa General Hospital, Tokyo Dental College. All study participants were briefed about the aims of the study and the parameters to be measured, and consent was obtained. The study was approved by the ethics committees of RIKEN, Tokyo University, and Ichikawa General Hospital. Sperm concentration and motility were measured using a computer-assisted image analyzer (C-Men, Compix, Cranberry Township, PA). Human semen was diluted twice with saline, layered on 5.0 mL of 20 mM HEPES buffered 90% isotonic Percoll (Cytiba, Uppsala, Sweden), and centrifuged at 400 × *g* for 22 min. The sperm in the sediment was recovered to yield 0.2 mL, and then introduced to the bottom of 2.0 mL of Hanks' solution to facilitate swim-up. The motile sperm in the upper layer were recovered.

### Preparation of SVA amplicon-seq

Genomic DNA was subjected to bisulfite-mediated C to U conversion using the MethylCode Bisulfite Conversion Kit (ThermoFisher Scientific), and then used as a template for PCR for 35 cycles with EpiTaq (Takara) using the following primers: SVA_1_Fw TTATTGTAATTTTTTTGTTTGATTTTTTTGTT TTAG. SVA_1_Rv AAAAAAACTCCTCACATCCCAAAC SVA_2_Fw TTAATGTTGTTTAGGTTGGAGTGT AGTG SVA_2_Rv CAAAAAAAACTCCTCACTTCCCAATA. SVA_3_Fw TTTGGGAGGTGTATTTAATAGTTT ATTGAGAA SVA_3_Rv TAAACAAAAATCTCTAATTTTCCTAAACAAAAAACC. The PCR products from three sets of primers were combined, purified using a MinElute PCR Purification Kit (QIAGEN), and fragmented using Picoruptor (Diagenode) for 10 cycles of 30 s on and 30 s off. Then, the amplicon-seq library was constructed using KAPA LTP Library Preparation Kits (KAPA BIOSYSTEMS) and SeqCap Adapter Kit A (Roche). The amplicon-seq libraries were sequenced on a HiSeq X platform (Illumina).

### WGBS and amplicon-seq analysis

We used the hg19 version of the human genome for NGS analysis because the predicted KRAB-ZFP peaks were derived from this version. Using the newest version of the human genome (GRCh38) did not significantly affect the conclusions. The following publicly available WGBS data were used in this study: hPGCs (SRP050499), sperm (SRP028572, ERP117337, JGAS00000000006), and adult tissues (IHEC data portal). For the IHEC data, we used processed data for our analysis.

- Quality control, read mapping, and DNA methylation calculation

Low-quality bases and adaptor sequences were trimmed using Trim Galore version 0.3.7 (http://www.bioinformatics.babraham.ac.uk/projects/trim_galore/). For WGBS data from hPGCs, the first nine bases were further trimmed. The trimmed reads were aligned to the hg19 genome using Bismark

v0.14.1, with paired-end and non-directional mapping parameters (--non_directional) (*Krueger and Andrews, 2011*). The unmapped reads after paired-end mapping were re-aligned to the same reference in single-end mode. We validated that this mapping mode only reported uniquely mapped reads. The methylation level of each CpG site was calculated as follows: (number of methylated reads/ number of total reads). Only CpG sites with at least five reads were used for all analyses. Only nearly full-length retroelements, whose length is 90% or more of the length of the consensus sequence of each subtype, were used for DNA methylation analysis of retroelements. We also included solo-LTR transposons in the DNA methylation analysis if they also possessed more than 90% of the consensus LTR sequence. Retroelement information was obtained from the UCSC Genome Browser (http:// genome.ucsc.edu/). For the DNA methylation analysis of retroelements, we used retroelements containing at least 10 CpG sites with a read depth of at least five reads. The methylation level of each retroelement copy was calculated by averaging the methylation levels of CpG sites within the copy.

Classification of retroelement copy according to DNA methylation levels.

Retroelement copies were classified according to their DNA methylation levels as follows: low < 20%, 20% ≦ medium < 60%, high ≧ 60%.

- Association of KRAB-ZFP peaks, binding motifs, and retroelements.

We obtained the peak regions of 250 KRAB-ZFP, which were previously reported (*Helleboid et al., 2019*; *Imbeault et al., 2017*), from the gene expression omnibus GSE78099 and GSE120539. Overlap of the KRAB-ZFP peak and retroelement copy was investigated using bedtools v2.15.0 (*Quinlan and Hall, 2010*). The binding motif of each KRAB-ZFP was predicted by the findMotifsGenome.pl program in Homer v4.8.3 (*Heinz et al., 2010*). The KRAB-ZFP-binding motifs along retroelement copies were searched using FIMO (*Grant et al., 2011*). We used predicted motif sites with a q-value of 0.00005 or less for ZNF257/ZNF28/ZNF850 and with a q-value of 0.05 or less for ZNF93 and ZNF649 in this study.

- DMR identification

DMR candidates were identified using the 'Commet' command in BisulFighter (*Saito et al., 2014*). To enhance the confidence of DMR call, we calculated the average methylation levels of the candidates using CpG sites with ≥5 reads in both sperm donors, and among the candidates, those containing ≥10 successive analyzable CpG sites and showing a ≥40% methylation difference were determined as DMRs.

## Phylogenetic analysis of retroelement copies

The evolutionary history was inferred using the maximum likelihood method based on the Tamura-Nei model (*Tamura and Nei, 1993*). The initial tree(s) for the heuristic search were obtained by applying the neighbor-joining method to a matrix of pairwise distances estimated using the maximum composite likelihood approach. The tree was drawn to scale, with branch lengths measured as the number of substitutions per site. There were 10,153 positions in the final dataset. Evolutionary analyses were conducted using MEGA6 (*Tamura et al., 2013*).

## Calculation of read mappability of each retroelement copy

We generated 100 bp reads from each position along the retroelement copy and aligned the simulated reads to the human genome using Bowtie with –m 1 or Bismark. Then, the mappability of each copy was calculated by dividing the number of properly mapped reads by the total number of reads derived from each copy.

## RNA-seq analysis

We reanalyzed previously reported single-cell RNA-seq data from the testes (*Sohni et al., 2019*), hPGCs, and somatic cells next to hPGCs (*Guo et al., 2015*). Read count data of genes and cell type annotation of each cell were obtained from the Gene Expression Omnibus under accession numbers GSE124263 and GSE63818. Reads per million mapped reads (RPM) for the genes were calculated for each cell. We used the average RPM of spermatogonial stem cells 2 (*Figure 5C*). We also reanalyzed previously reported RNA-seq data from undifferentiated spermatogonia (*Tan et al., 2020*), which was deposited in the Gene Expression Omnibus under accession number GSE144085. Low-quality bases and adaptor sequences were trimmed using Trim Galore version 0.3.7. Then, trimmed reads

were aligned to the hg19 genome using Bowtie v0.12.7 with -m 1 to remove multiple mapped reads. Enrichment of RNA-seq reads around SVAs was visualized using ngsplot (*Shen et al., 2014*).

### ChIP-seq analysis

We reanalyzed previously reported KRAB-ZFP ChIP-exo data (*Helleboid et al., 2019*; *Imbeault et al., 2017*), which were deposited in the Sequence Read Archive SRP070561 and SRP162756. Low-quality bases and adaptor sequences were trimmed using the Trim Galore version 0.3.7. Then, trimmed reads were aligned to the hg19 genome using Bowtie v0.12.7 with -m 1 to avoid multiple mapped reads. Enrichment of ChIP-exo reads around retroelements was visualized using ngsplot (*Shen et al., 2014*).

### Visualization of NGS data

The Integrative Genomics Viewer (*Robinson et al., 2011*) was used to visualize the NGS data. Enrichment of RNA-seq reads and KRAB-ZFPs was visualized using ngsplot (*Shen et al., 2014*). Scatter plot and violin plot analyses were performed using the ggplot2 package in R.

### Data access

All reads from amplicon-seq in this study have been submitted to the Gene Expression Omnibus under accession number GSE174562.

## Acknowledgements

We thank the staff of the Support Unit for Bio-Material Analysis (BMA) at the RIKEN Center for Brain Science (CBS) Research Resources Division (RRD) for NGS library construction. This research was supported by the Special Postdoctoral Researcher (SPDR) Program of RIKEN to KF, the Japan Ministry of Education, Culture, Sports, Science, and Technology Grant-in-Aid for Scientific Research (18H05530, 18H03991) to YS, and RIKEN internal research funds to YS.

## Additional information

### Funding

| Funder | Grant reference number | Author |
|--------|------------------------|--------|
| Japan Society for the Promotion of Science | 18H05530 | Yoichi Shinkai |
| Japan Society for the Promotion of Science | 18H03991 | Yoichi Shinkai |
| RIKEN | SPDR | Kei Fukuda |

The funders had no role in study design, data collection and interpretation, or the decision to submit the work for publication.

### Author contributions

Kei Fukuda, Conceptualization, Data curation, Formal analysis, Investigation, Validation, Writing - original draft, Writing - review and editing; Yoshinori Makino, Satoru Kaneko, Yuki Okada, Resources; Chikako Shimura, Investigation; Kenji Ichiyanagi, Conceptualization; Yoichi Shinkai, Conceptualization, Funding acquisition, Resources, Supervision, Writing - original draft, Writing - review and editing

### Author ORCIDs

Kei Fukuda http://orcid.org/0000-0002-4111-9409
Yoichi Shinkai http://orcid.org/0000-0002-6051-2484

### Ethics

Human subjects: This study was approved by the ethics committees of RIKEN, Tokyo University, and Ichikawa General Hospital.All study participants were briefed about the aims of the study and the parameters to be measured, and consent was obtained.

Decision letter and Author response
Decision letter https://doi.org/10.7554/eLife.76822.sa1
Author response https://doi.org/10.7554/eLife.76822.sa2

## Additional files

### Supplementary files
• Supplementary file 1. List of full-length L1/SVA/LTR12.

• Supplementary file 2. Statistical analysis of DNA methylation levels of repeat elements in human primordial germ cells (PGCs) (related to *Figure 1C–E*).

• Transparent reporting form

### Data availability

All reads from amplicon-seq in this study have been submitted to the Gene Expression Omnibus under accession number GSE174562.

The following dataset was generated:

| Author(s) | Year | Dataset title | Dataset URL | Database and Identifier |
|---|---|---|---|---|
| Fukuda K, Shinkai Y | 2021 | Amplicon-seq of SVA methylation in human sperm | http://www.ncbi.nlm.nih.gov/geo/query/acc.cgi?acc=GSE174562 | NCBI Gene Expression Omnibus, GSE174562 |

The following previously published datasets were used:

| Author(s) | Year | Dataset title | Dataset URL | Database and Identifier |
|---|---|---|---|---|
| Imbeault M, Helleboid PY, Trono D | 2017 | ChIP-exo of human KRAB-ZNFs transduced in HEK 293T cells and KAP1 in hES H1 cells | https://www.ncbi.nlm.nih.gov/geo/query/acc.cgi?acc=GSE78099 | NCBI Gene Expression Omnibus, GSE78099 |
| Guo F, Guo H, Li L, Tang F | 2015 | The Transcriptome and DNA Methylome Landscapes of Human Primordial Germ Cells | https://www.ncbi.nlm.nih.gov/geo/query/acc.cgi?acc=GSE63818 | NCBI Gene Expression Omnibus, GSE63818 |
| Low DH, Hammoud SS | 2014 | Chromatin and Transcription Transitions of Mammalian Adult Germline Stem Cells and Spermatogenesis | https://www.ncbi.nlm.nih.gov/geo/query/acc.cgi?acc=GSE49624 | NCBI Gene Expression Omnibus, GSE49624 |
| Okae H, Chiba H, Hiura H, Hamada H, Sato A, Utsunomiya T, Kikuchi H, Yoshida H, Tanaka A, Suyama M, Arima T | 2014 | Genome-Wide Analysis of DNA Methylation Dynamics during Early Human Development | https://humandbs.biosciencedbc.jp/hum0009-v1 | the Japanese Genotype-phenotype Archive, JGAS00000000006 |
| Sohni A, Tan K, Song H, Burow D, Wilkinson MF | 2018 | Neonatal and adult human testis defined at the single-cell level | https://www.ncbi.nlm.nih.gov/geo/query/acc.cgi?acc=GSE124263 | NCBI Gene Expression Omnibus, GSE124263 |

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
