## [Editor Report]

Retrotransposons undergo massive reprogramming of their methylation states during germ cell development, but some elements are immune to this remodeling. This manuscript explores the contribution of binding motifs for KRAB-Zinc Finger Proteins (KZFPs) and position towards genes to explain the variable methylation dynamics of different retrotransposon families, namely L1, SVA and LTR12, as well as potential inter-individual variation during male germ cell development in humans, using an integrative analyses of available sequencing datasets. By bringing insights into the complex regulation of retrotransposons, it could be of particular interest to the epigenetics community.

---

## [Decision Letter]

**Decision letter after peer review:**

Thank you for submitting your article "Transcriptional states of retroelement-inserted regions and specific KRAB zinc finger protein association are correlated with DNA methylation of retroelements in human male germ cells" for consideration by *eLife*. Your article has been reviewed by 2 peer reviewers, and the evaluation has been overseen by a Reviewing Editor and Molly Przeworski as the Senior Editor. The following individuals involved in review of your submission have agreed to reveal their identity: Michael Imbeault (Reviewer #1); Geoffrey Faulkner (Reviewer #2).

Essential revisions:

The reviewers agreed the study was well conducted and although was initially done on a limited number of available samples, the important observation regarding the inter-individual variability of SVA methylation in sperm was replicated in an additional dataset with more donors and further confirmed by targeted amplicon sequencing. The conclusions regarding the DNA methylation dependency towards specific KZFPs is not novel, although it was extended here to the context of primordial germ cells. The most intriguing conclusion is certainly the description of inter-individual differences in SVA methylation in the sperm of different donors. A link with being positioned inside a highly transcribed gene and in reverse orientation towards the host gene was drawn regarding the propensity of being methylated for an SVA, but the explanation for this is very uncertain. Results appear as preliminary on this matter.

Please address the following points in a revised version of your manuscript, and answer the individual comments and questions raised by the reviewers in a detailed rebuttal letter, with possible corrections and additional figures to be included in the manuscript.

1 – Integrate all the KZFP datasets that are available in GSE120539 (but that were not published) in your analysis

2 – Is it know whether interindividual differences in SVA methylation are also found in somatic tissues, or is it a specific feature of sperm DNA? If this has not been described, please analyze available WGBS datasets focusing on one tissue of several donors, or use your amplicon sequencing strategy. This would be an important addition to the biological meaning of such differences, whether this variability also occurs during embryonic reprogramming, not only germ cell reprogramming.

3 – Provide more information as to the methodology for full length retroelement analysis and ages of sperm donors

4 – Please show screen shots of individual loci that follow the stated correlation of DNA methylation and KZFP binding

5 – Review all sentences with over statements, as outlined by reviewer #1.

Additionally, please correct the following points:

– Line 304: "… subsequently, MIWI2-interacting protein SPOCD1 recruits the chromatin remodeling complex DNMT3A and DNMT3L to…". This sentence is wrong twice: (1) DNMT3A-DNMT3L does not have chromatin remodeling ability per se, it is a de novo DNA methylation complex, and (2) DNMT3C methylates piRNA-targeted retroelements in mice, not DNMT3A (see Barau et al., 2016). Please correct as you are referring here to the process that has been described in mice.

– The link between intragenic position of SVAs and their ability to undergo methylation by transcriptional readthrough/H3K36 methylation is interesting, as it means it would happen in a piRNA-independent manner. However, it does not explain why only the SVA elements with a reverse orientation to the host gene would be affected. Please report this.

*Reviewer #1 (Recommendations for the authors):*

I am a researcher active in the field of KRAB zinc finger proteins for about 12 years. In my opinion, the science in this manuscript is high quality and the findings as novel as they are interesting.

I have a few small recommendations to improve some sections of the manuscript and would appreciate if they can be implemented before publication.

Early in the manuscript it is stated that "we focused on full-length copies of retroelements to analyze 85 DNA methylation for at least 30 copies.". What did you consider as 'full-length', especially for LTR-containing retroviruses? Did you analyze only copies containing the internal part, discarding solo-LTRs? I could not find details describing this in the methods section. Also, this decision is probably biasing the analysis toward younger elements – not a problem in itself, but it should be stated clearly.

With very young elements (L1Hs) the mappability is probably not very good with 100bp non-paired end reads. Could you provide a supplemental table with average mappability per family of transposons, or as a supplemental figure as a violinplot of mappability per family.

There's more data of KZFP that is available that was not included – notably at GEO accession GSE120539 from the Trono lab – these are from the same experimental series as the ones published initially but didn't make it through analysis before publication – it would be great if you can include them.

For figure 2, I would like to see statistics of enrichment (p-value) for overlaps with specific KZFPs / families of transposons and DNA methylation categories.

Figure 2A – a heatmap is not the best visualization here – it could be a simple sorted barchart of hits zoomed in on the first few members with the highest scores. Same comment for figure 3H.

Review all sentences that are related to conclusions to avoid overly strong wording, considering that most findings of this manuscript are purely correlative and causation has not been demonstrated. As an example (out of many): "Therefore, SVAs are methylated during spermatogenesis if these are inserted into actively transcribed genes." – this is too strong, you might want to add 'suggest, might, potentially'.

Please discuss somewhere in the manuscript the potential for multiple KZFPs binding the same elements having a concerted effect on the elements.

Finally, I would like to see the age distribution of the sperm donors, and some analysis to see if variability is correlating with age in any way.

*Reviewer #2 (Recommendations for the authors):*

My two major reservations are the claims around inter-individual variability being difficult to distinguish from technical variablity, which I don't have a reasonable suggestion for how to address, and the first specific point above, namely that uniquely mapped WGBS reads are unlikely to measure methylation in the core VNTR region of an SVA (or the L1HS 5'UTR CpG island). The authors could address this point by showing a composite profile of WGBS coverage and methylation levels compared to L1/SVA consensus sequences, showing the inner parts of L1 and SVA. They could also show examples of individual loci that follow the stated patterns of DNA methylation and ZFP binding that supports a correlation between the two. Another option would be to do nanopore long-read sequencing, which obviously would take time and substantial resources, but would provide a comprehensive picture of the situation. Note that this issue affects some of the more high profile mouse retrotransposons, such as IAP, to a much lesser degree because their LTRs are more accessible to WGBS.

I think the figure legends for Figure 2E and Figure 2F are swapped.

---

## [Author Response]

1 – Integrate all the KZFP datasets that are available in GSE120539 (but that were not published) in your analysis

Thanks for your suggestion. We reanalyzed KZFP peak data from GSE120539 and found that ZNF850 was enriched in highly methylated LTR12C copies in hPGCs. We included the analysis of ZNF850 in Supplementary Figure S4.

2 – Is it know whether interindividual differences in SVA methylation are also found in somatic tissues, or is it a specific feature of sperm DNA? If this has not been described, please analyze available WGBS datasets focusing on one tissue of several donors, or use your amplicon sequencing strategy. This would be an important addition to the biological meaning of such differences, whether this variability also occurs during embryonic reprogramming, not only germ cell reprogramming.

Thanks for the suggestion. This is also important point.

To address this issue, we reanalyzed IHEC DNA methylation data from adult skeletal muscle (15 samples) and CD4-positive helper T cells (18 samples). As all SVA groups showed high DNA methylation in all individuals tested, inter-individual variation of SVA DNA methylation in sperm would be cancelled during development (Supplementary Figure S5A, B). We stated this result in the text.

“In contrast to the inter-individual epigenetic variation of SVAs in sperm, a reanalysis of WGBS data of adult skeletal muscle from 15 individuals and of helper CD4-positive T cells from 18 individuals, which was deposited in the International Human Epigenome Consortium (IHEC) portal, showed high DNA methylation of SVAs in all individuals (Bujold et al., 2016) (Figure 6—figure supplement 1A, B). Thus, inter-individual variations in DNA methylation of SVAs in sperm are essentially canceled during development.” P15. line 276-281.

3 – Provide more information as to the methodology for full length retroelement analysis and ages of sperm donors

We added following sentences in the method section for the full length retroelement analysis.

“Only nearly full-length retroelements, whose length is 90% or more of the length of the consensus sequence of each subtype, were used for DNA methylation analysis of retroelements. We also included solo-LTR transposons in the DNA methylation analysis if they also possessed more than 90% of the consensus LTR sequence.” P34, line 606-609.

The ages of sperm donors used in Hammoud *et al.,* data were described in the manuscript. On the other hand, we could not access clinical information of our samples for amplicon-seq, so age data for Okae *et al.*, was not available.

4 – Please show screen shots of individual loci that follow the stated correlation of DNA methylation and KZFP binding

I added the representative loci in Figure S1J, S4G, 3H.

5 – Review all sentences with over statements, as outlined by reviewer #1.

We revised following sentences to be not over described as possible as we can.

Therefore, SVAs are methylated during spermatogenesis if these are inserted into actively transcribed genes.

“Therefore, SVAs located in actively transcribed regions in the antisense orientation are efficiently subjected to de novo DNA methylation during spermatogenesis.” p14, line 245-246

The data indicate that the effectiveness of the de novo DNA methylation varies among individuals.

“These results implicate the possibility that SVAs acquire DNA methylation during DNA methylation via transcription-directed machinery, and that the effectiveness of de novo DNA methylation varies among individuals.” p15, line 255-257

SVAs constitute a major source of inter-individual epigenetic variation in sperm

“SVAs are a potential source of inter-individual epigenetic variation in sperm”

p15, line 259

In this study, we showed that KRAB-ZFPs are associated with the DNA demethylation resistance of retroelements in hPGCs.

“Thus, the involvement of other KRAB-ZFPs in DNA demethylation resistance of retroelements in hPGCs is possible.” p19, line 330-331

SVAs are subjected to de novo DNA methylation by the transcription-directed DNA methylation machinery.

“the SVAs located in transcription-active regions in the antisense orientation are prone to methylation during spermatogenesis, which implies that the transcription-directed DNA methylation machinery might contribute to de novo DNA methylation of SVAs in male germ cells.” p18, line 302-304

Our data showed that the acquisition of DNA methylation of SVAs during spermatogenesis was regulated by the inserted regions and not by the nucleotide sequence. SVAs inserted in transcriptionally active regions in the antisense direction were targeted by de novo DNA methylation during spermatogenesis.

“Our data showed that the acquisition of DNA methylation of SVAs during spermatogenesis correlated with the inserted regions and not with the nucleotide sequence. SVAs inserted in transcriptionally active regions in the antisense direction are efficiently targeted for de novo DNA methylation during spermatogenesis.” P20, line 354- p21, 357

Additionally, please correct the following points:– Line 304: "… subsequently, MIWI2-interacting protein SPOCD1 recruits the chromatin remodeling complex DNMT3A and DNMT3L to…". This sentence is wrong twice: (1) DNMT3A-DNMT3L does not have chromatin remodeling ability per se, it is a de novo DNA methylation complex, and (2) DNMT3C methylates piRNA-targeted retroelements in mice, not DNMT3A (see Barau et al., 2016). Please correct as you are referring here to the process that has been described in mice.

Thanks for the suggestion. We changed the sentences as bellow.

“In mouse spermatogenesis, MIWI2 binds piRNAs and is recruited to the nascent transcribed regions that are complementary to piRNAs (Watanabe et al., 2018). Subsequently, MIWI2 interacting protein SPOCD1, which forms a complex with DNMT3A and DNMT3L, and potentially with a rodent-specific DNA methyltransferase DNMT3C (Barau et al., 2016), induces DNA methylation on transposons (Zoch et al., 2020)..” P21, line 357- 361.

– The link between intragenic position of SVAs and their ability to undergo methylation by transcriptional readthrough/H3K36 methylation is interesting, as it means it would happen in a piRNA-independent manner. However, it does not explain why only the SVA elements with a reverse orientation to the host gene would be affected. Please report this.

Thank you for this comment. We described this issue as bellow.

“Furthermore, antisense RNAs embedded within protein-coding genes are selectively silenced by H3K36 methyltransferase SET2 in *Saccharomyces cerevisiae* (Venkatesh, Li, Gogol, and Workman, 2016). Thus, the machinery for gene body DNA methylation regulated by SETD2 is also a candidate for the de novo DNA methylation of SVAs during spermatogenesis.” p22, line 379-383.

Reviewer #1 (Recommendations for the authors):I am a researcher active in the field of KRAB zinc finger proteins for about 12 years. In my opinion, the science in this manuscript is high quality and the findings as novel as they are interesting.I have a few small recommendations to improve some sections of the manuscript and would appreciate if they can be implemented before publication.Early in the manuscript it is stated that "we focused on full-length copies of retroelements to analyze 85 DNA methylation for at least 30 copies.". What did you consider as 'full-length', especially for LTR-containing retroviruses? Did you analyze only copies containing the internal part, discarding solo-LTRs? I could not find details describing this in the methods section. Also, this decision is probably biasing the analysis toward younger elements – not a problem in itself, but it should be stated clearly.

Thank you for your valuable and most KRAB-ZFP expert comments on our manuscript. As already pointed out, we also included solo-LTR copies in our DNA methylation analysis, if they possess more than 90% of the consensus LTR sequence. Length of consensus sequence was obtained from RepeatMasker data in UCSC genome browser. We described this in the Method section.

In addition, we stated our analysis was biased toward younger elements in Discussion section as bellow.

“Because we targeted full-length transposons, our analysis was biased towards relatively young transposons. Thus, it is possible that some fragmented older transposons may also be resistant to DNA demethylation in hPGCs.” P17, line 309-311.

With very young elements (L1Hs) the mappability is probably not very good with 100bp non-paired end reads. Could you provide a supplemental table with average mappability per family of transposons, or as a supplemental figure as a violinplot of mappability per family.

Thanks for the suggestion. To investigate mappability of WGBS read, we generated 100-bp sequences from each full-length copy *in silico* and mapped the reads to human genome by bismark. As the reviewer pointed out, the younger transposon, the lower mapping efficiency. This result was included in Supplementary Figure S1A.

There's more data of KZFP that is available that was not included – notably at GEO accession GSE120539 from the Trono lab – these are from the same experimental series as the ones published initially but didn't make it through analysis before publication – it would be great if you can include them.

Thanks for your good suggestion. We reanalyzed KZFP peak data from GSE120539 and found that ZNF850 was enriched in highly methylated LTR12C copies in hPGCs. We included the analysis of ZNF850 in Supplementary Figure S4. As we found KRAB-ZFP associated with LTR12 methylation, we removed old Figure S4 and Figure 3H.

For figure 2, I would like to see statistics of enrichment (p-value) for overlaps with specific KZFPs / families of transposons and DNA methylation categories.

Expected and observed numbers of SVA_A and LTR12C that overlap with KRAB-ZNF binding peaks were calculated and *P-*value for enrichment of retroelement in KRAB-ZNF binding peaks was calculated by *prop.*test by R. We included these data in Supplementary Figure S1C.

Figure 2A – a heatmap is not the best visualization here – it could be a simple sorted barchart of hits zoomed in on the first few members with the highest scores. Same comment for figure 3H.

Thanks for the comment.

We changed the heatmap in Figure 2A and Figure 3H to scatter plot.

Review all sentences that are related to conclusions to avoid overly strong wording, considering that most findings of this manuscript are purely correlative and causation has not been demonstrated. As an example (out of many): "Therefore, SVAs are methylated during spermatogenesis if these are inserted into actively transcribed genes." – this is too strong, you might want to add 'suggest, might, potentially'.

Thanks for the suggestion. We revised following sentences to be not over described as possible as we can.

Therefore, SVAs are methylated during spermatogenesis if these are inserted into actively transcribed genes.

“Therefore, SVAs located in actively transcribed regions in the antisense orientation are efficiently subjected to de novo DNA methylation during spermatogenesis.” p14, line 245-246

The data indicate that the effectiveness of the de novo DNA methylation varies among individuals.

“These results implicate the possibility that SVAs acquire DNA methylation during DNA methylation via transcription-directed machinery, and that the effectiveness of de novo DNA methylation varies among individuals.” p15, line 255-257

SVAs constitute a major source of inter-individual epigenetic variation in sperm

“SVAs are a potential source of inter-individual epigenetic variation in sperm”

p15, line 259

In this study, we showed that KRAB-ZFPs are associated with the DNA demethylation resistance of retroelements in hPGCs.

“Thus, the involvement of other KRAB-ZFPs in DNA demethylation resistance of retroelements in hPGCs is possible.” p19, line 330-331

SVAs are subjected to de novo DNA methylation by the transcription-directed DNA methylation machinery.

“the SVAs located in transcription-active regions in the antisense orientation are prone to methylation during spermatogenesis, which implies that the transcription-directed DNA methylation machinery might contribute to de novo DNA methylation of SVAs in male germ cells.” p18, line 302-304

Our data showed that the acquisition of DNA methylation of SVAs during spermatogenesis was regulated by the inserted regions and not by the nucleotide sequence. SVAs inserted in transcriptionally active regions in the antisense direction were targeted by de novo DNA methylation during spermatogenesis.

“Our data showed that the acquisition of DNA methylation of SVAs during spermatogenesis correlated with the inserted regions and not with the nucleotide sequence. SVAs inserted in transcriptionally active regions in the antisense direction are efficiently targeted for de novo DNA methylation during spermatogenesis.” P20, line 354- p21, 357

Please discuss somewhere in the manuscript the potential for multiple KZFPs binding the same elements having a concerted effect on the elements.

We added the following sentence in the Discussion section.

“In hPGCs, multiple KRAB-ZNPs were correlated with DNA demethylation resistance in the same retroelements, which may contribute to more robust or cooperative retroelement suppression.” P18, line 319- p19, line 321

Finally, I would like to see the age distribution of the sperm donors, and some analysis to see if variability is correlating with age in any way.

The ages of sperm donors used in Hammoud *et al.,* data were described in the manuscript (Donor 1 -32 yo, Donor2 -37 yo). On the other hand, we could not access clinical information of our samples for amplicon-seq, so age data for Okae *et al.,* was not available. To investigate the correlation of SVA methylation with age, we further analyzed publicly available sperm WGBS data from five healthy individuals and six oligozoospermic patients, which are deposited in the European Nucleotide Archive (ENA) under the accession number PRJEB34432 (Leitao et al., Clinical Epigenetics, 2020) and are linked to clinical information including age. We found that both healthy and patients showed variable SVA methylation and age was not associated with SVA methylation state. We also found that testosterone level in blood was negatively correlated with SVA methylation in sperm, implicating potential link between SVA methylation in sperm and physiological conditions. These data were included in Supplementary Figure S5C, D and described in the text as follow.

“Finally, to investigate whether inter-individual DNA methylation variations are associated with physiological or disease conditions, we reanalyzed publicly available WGBS data from five healthy donors and six oligozoospermic patients (European Nucleotide Archive (ENA) under the accession number PRJEB34432) (Leitao et al., 2020). The disease condition was not associated with inter-individual DNA methylation variations in SVAs, because both healthy donors and oligozoospermic patients showed inter-individual variations of DNA methylation in “High and Low” SVA_B–F copies (Figure 6—figure supplement 1C). On the other hand, a comparison of various physiological conditions between highly methylated individuals and lowly methylated ones (median methylation levels of “High and Low” > 50% vs < 50%) revealed that blood testosterone levels were significantly higher in lowly methylated individuals than in highly methylated ones (Figure 6—figure supplement 1D). However, prolactin, follicle stimulating hormone (FSH), luteinizing hormone (LH), sex hormone-binding globulin (SHBG) blood levels and age were not significantly different between the two groups (Figure 6—figure supplement 1D). Although further validation of this correlation is required, DNA methylation of SVAs in sperm may be associated with physiological conditions.” P16, line 282- P17, line 296.

Reviewer #2 (Recommendations for the authors):My two major reservations are the claims around inter-individual variability being difficult to distinguish from technical variablity, which I don't have a reasonable suggestion for how to address, and the first specific point above, namely that uniquely mapped WGBS reads are unlikely to measure methylation in the core VNTR region of an SVA (or the L1HS 5'UTR CpG island). The authors could address this point by showing a composite profile of WGBS coverage and methylation levels compared to L1/SVA consensus sequences, showing the inner parts of L1 and SVA. They could also show examples of individual loci that follow the stated patterns of DNA methylation and ZFP binding that supports a correlation between the two. Another option would be to do nanopore long-read sequencing, which obviously would take time and substantial resources, but would provide a comprehensive picture of the situation. Note that this issue affects some of the more high profile mouse retrotransposons, such as IAP, to a much lesser degree because their LTRs are more accessible to WGBS.

Thanks for your valuable comment/suggestion. As the reviewer pointing out, it is difficult to measure DNA methylation in core VNTR regions of a younger SVA. But we were able to do it in the oldest SVA type, SVA_A. We showed the DNA methylation levels in VNTR regions of SVA_A and overlap with KRAB-ZFP peaks in new Supplementary Figure S1H. We also showed examples of individual loci showing correlation of DNA methylation in hPGCs and KRAB ZFP association in SVA_A and L1PA4 in new Supplementary Figure S1I,J and 3H. And we described about this issue as bellow.

“We also confirmed that DNA methylation levels within the VNTR were correlated with ZNF257 or ZNF28 association (Figure 2—figure supplement 1G, H).” p9, 149-151

“We also confirmed high DNA methylation in the ZNF649 binding motifs at individual loci (Figure 3H).” p11, 185-186

I think the figure legends for Figure 2E and Figure 2F are swapped.

Thank you. We fixed it.